# Primary Ciliary Dyskinesia and Retinitis Pigmentosa: Novel *RPGR* Variant and Possible Modifier Gene

**DOI:** 10.3390/cells13060524

**Published:** 2024-03-16

**Authors:** Noelia Baz-Redón, Laura Sánchez-Bellver, Mónica Fernández-Cancio, Sandra Rovira-Amigo, Thomas Burgoyne, Rai Ranjit, Virginia Aquino, Noemí Toro-Barrios, Rosario Carmona, Eva Polverino, Maria Cols, Antonio Moreno-Galdó, Núria Camats-Tarruella, Gemma Marfany

**Affiliations:** 1Growth and Development Research Group, Vall d’Hebron Research Institute (VHIR), Hospital Universitari Vall d’Hebron, 08035 Barcelona, Spain; noelia.baz@vhir.org (N.B.-R.); monica.fernandez.cancio@vhir.org (M.F.-C.); sandra.rovira@vallhebron.cat (S.R.-A.); antonio.moreno@vallhebron.cat (A.M.-G.); 2Centro de Investigación Biomédica en Red de Enfermedades Raras (CIBERER), Instituto de Salud Carlos III, 28029 Madrid, Spain; lsanchezbel@ub.edu (L.S.-B.); rosariom.carmona@juntadeandalucia.es (R.C.); gmarfany@ub.edu (G.M.); 3Departament de Genètica, Microbiologia i Estadística, Universitat de Barcelona, 08028 Barcelona, Spain; 4Department of Paediatrics, Vall d’Hebron Hospital Universitari, Vall d’Hebron Barcelona Hospital Campus, 08035 Barcelona, Spain; 5Royal Brompton Hospital, Guy’s and St Thomas’ NHS Foundation Trust, London SW3 6NP, UK; t.burgoyne@ucl.ac.uk (T.B.); ranjit.k.rai@durham.ac.uk (R.R.); 6Institute of Ophthalmology, University College London, London EC1V 9EL, UK; 7Plataforma Andaluza de Medicina Computacional, Fundación Pública Andaluza Progreso y Salud, 41092 Sevilla, Spain; virginia.aquino@juntadeandalucia.es (V.A.); noemi.toro@juntadeandalucia.es (N.T.-B.); 8Pneumology Research Group, Vall d’Hebron Research Institute (VHIR), Hospital Universitari Vall d’Hebron, 08035 Barcelona, Spain; eva.polverino@vallhebron.cat; 9Pneumology Department, Vall d’Hebron Hospital Universitari, Vall d’Hebron Barcelona Hospital Campus, 08035 Barcelona, Spain; 10Centro de Investigación Biomédica en Red de Enfermedades Respiratorias (CIBERES), Instituto de Salud Carlos III, 28029 Madrid, Spain; 11Paediatric Pulmonology Department and Cystic Fibrosis Unit, Hospital Sant Joan de Déu, 08950 Esplugues de Llobregat, Spain; maria.cols@sjd.es; 12Department of Paediatrics, Obstetrics, Gynecology, Preventive Medicine and Public Health, Universitat Autònoma de Barcelona, 08193 Bellaterra, Spain; 13Institute of Biomedicine (IBUB-IRSJD), Universitat de Barcelona, 08028 Barcelona, Spain

**Keywords:** primary cilia, motile cilia, primary ciliary dyskinesia, retinitis pigmentosa, PCD, XLRP, *RPGR*, *CEP290*, modifier gene

## Abstract

We report a novel *RPGR* missense variant co-segregated with a familial X-linked retinitis pigmentosa (XLRP) case. The brothers were hemizygous for this variant, but only the proband presented with primary ciliary dyskinesia (PCD). Thus, we aimed to elucidate the role of the *RPGR* variant and other modifier genes in the phenotypic variability observed in the family and its impact on motile cilia. The pathogenicity of the variant on the RPGR protein was evaluated by in vitro studies transiently transfecting the mutated *RPGR* gene, and immunofluorescence analysis on nasal brushing samples. Whole-exome sequencing was conducted to identify potential modifier variants. In vitro studies showed that the mutated RPGR protein could not localise to the cilium and impaired cilium formation. Accordingly, RPGR was abnormally distributed in the siblings’ nasal brushing samples. In addition, a missense variant in *CEP290* was identified. The concurrent *RPGR* variant influenced ciliary mislocalisation of the protein. We provide a comprehensive characterisation of motile cilia in this XLRP family, with only the proband presenting PCD symptoms. The variant’s pathogenicity was confirmed, although it alone does not explain the respiratory symptoms. Finally, the *CEP290* gene may be a potential modifier for respiratory symptoms in patients with *RPGR* mutations.

## 1. Introduction

Primary ciliary dyskinesia (PCD) is an autosomal recessive disease (1/7500) caused by an alteration of the ciliary structure, which impairs mucociliary clearance [1,2,3]. Symptoms of PCD may include persistent wet cough from early infancy, recurrent respiratory infections, bronchiectasis, chronic rhinosinusitis, persistent otitis media and associated hearing loss, male infertility, female subfertility, situs inversus in half of PCD patients [1,2,4] and heterotaxy in 6–12% of cases [5]. Although most genetic defects associated with PCD are inherited in an autosomal recessive manner, it has also been described to overlap with X-linked retinitis pigmentosa (XLRP) caused by mutations in the *RPGR* gene, which encodes a common component of motile and primary cilia. Retinitis pigmentosa (RP) is an inherited retinal degenerative disease (rod–cone dystrophy), resulting in progressive vision loss that eventually leads to blindness [6]. RP is a highly heterogeneous genetic disorder, with the X-linked form (XLRP, 6–20%) being one of the most severe [7]. The retinitis pigmentosa GTPase regulator (*RPGR*) gene was the first XLRP-causing gene identified [8] and is responsible for about 80% of the X-linked cases [9]. *RPGR* is also a major gene for X-linked cone/cone–rod dystrophy [10].

Although some authors reported no respiratory symptoms in XLRP families and *RPGR* variants [11,12], other authors described the presence of PCD symptoms with motile ciliary defects in a variable number of males in XLRP families, indicating the incomplete penetrance of respiratory symptoms in those cases [13,14,15,16,17,18,19]. McCray et al. [20] observed a high percentage of XLRP patients with ciliary orientation abnormalities and disorganised ciliary beat, but without clinical consequences [20]. In the previously reported families with respiratory symptoms, all pathogenic *RPGR* variants were missense, frameshift or causative of aberrant splicing [13,14,15,16,17]. Irrespective of the predicted variant outcome, respiratory manifestations were similar among all patients, including bronchiectasis [14,15] and serous otitis and hearing loss [15,16,17]. No case presented with laterality defects [13,14,15,16].

The *RPGR* gene is located on the X chromosome (Xp11.4) and produces at least 20 isoforms, resulting from complex alternative splicing events [21]. The *RPGR* constitutive isoform is RPGR_Ex1-19_, spanning from exon 1 to exon 19, and is expressed in several tissues, including lung, kidney, testis and brain [21,22,23]. The subcellular localisation of the protein encoded by this isoform is the transition zone (TZ) of motile and primary cilia [22,24]. The amino-terminal sequence (exons 3–10) includes an RCC1-like domain (RLD) that is responsible for protein–protein interactions [21,22,24,25]. The localisation and interacting partners of the RPGR protein support its role in ciliary function and intraflagellar trafficking of phototransduction proteins [16,21].

In this study, a novel *RPGR* missense variant was described as the cause of XLRP in a family, with PCD symptoms only in the proband. His brother also carried the variant but presented only with RP. The aims of this study were: (a) to characterise the motile cilia defects and the pathogenicity of the *RPGR* variant in this family; and (b) to identify modifier factors contributing to the variability in PCD symptoms in the RP and PCD syndrome.

## 2. Materials and Methods

Material and methods are extensively described in the Appendix A.

### 2.1. Patients

We present a family diagnosed with RP, referred to our pulmonary section because of respiratory PCD symptoms in the proband. The patient and his two siblings and parents were studied within the framework of a research project approved by the Clinical Research Ethics Committee (CEIC) of HUVH (PR(AMI)148/2016). We obtained written informed consents from all family members included in the study.

### 2.2. Primary Ciliary Dyskinesia Diagnostic Evaluation

Following the European Respiratory Society guidelines for PCD diagnosis [26] and in-house optimised protocols, we evaluated the ciliary function and structure in the proband, siblings and mother with a clinical symptoms questionnaire and PICADAR (PrImary CiliARy DyskinesiA Rule) score [27], nasal nitric oxide (nNO) screening test [28], high-speed video-microscopy analysis (HSVM) [29], immunofluorescence (IF) [30] and transmission electron microscopy (TEM) analyses, and genetics [31] (Appendix A, [32,33,34,35,36,37]).

### 2.3. In Silico Studies of the Candidate Variant

In silico studies involved the analysis of the amino acid conservation through evolution using UCSC Genome Browser (http://genome.ucsc.edu, accessed on 27 February 2023) [38], the prediction of protein stability changes caused by the variant using the I-Mutant2.0 website tool (https://folding.biofold.org/i-mutant/i-mutant2.0.html, accessed on 27 February 2023) and the effect of the variant in the RPGR protein structure modelled with PyMOL (The PyMOL Molecular Graphics System, Version 2.0 Schrödinger, LLC.) in the tertiary structure of RPGR RCC1 domain from AlphaFold [39,40].

### 2.4. In Vitro Functional Studies

In vitro studies were performed to confirm the pathogenicity of the identified *RPGR* variant in constructs expressing the fusion protein GFP–RPGR_Ex1-19_ [41]. After 48 h of transient transfection on the hTERT-RPE1 cell line, cells were analysed by IF using anti-γ-tubulin, anti-GFP and anti-acetylated-α-tubulin by confocal microscopy and ImageJ software (National Institutes of Health, Bethesda, MD, USA; Version 1.53c) for image processing and analysis (Appendix A).

### 2.5. Immunofluorescence Analyses of RPGR and CEP290 in Nasal Brushing Samples

Nasal respiratory epithelial cells were fixed and treated for IF with anti-γ-tubulin, anti-acetylated-α-tubulin and anti-RPGR or anti-CEP290. Samples were captured by confocal microscopy and analysed by ImageJ software (Appendix A).

### 2.6. Statistics

Data were analysed using GraphPad Prism software (GraphPad v9.0.1 Software Inc., San Diego, CA, USA). We used the non-parametric Kruskal–Wallis test for all data comparisons, considering significance with a *p*-value < 0.05.

## 3. Results

### 3.1. PCD Clinical Manifestation and Diagnosis

The proband (II-3) was previously diagnosed with XLRP, as were other members of his family (Figure 1a), because of blindness, but was referred to our paediatric pulmonary section due to respiratory symptoms and a PICADAR score of 6, suggesting PCD. His PCD-related symptoms were neonatal distress with admittance to a neonatal intensive care unit, chronic cough, recurrent otitis and pneumonia, hearing loss, chronic sinusitis, and diffuse bronchiectasis (Figure 1c). We evaluated the proband, siblings (II-1 and II-2) and mother (I-1) for PCD diagnoses from nasal brushing samples (Table 1). All family members presented with normal nNO values, and normal or close-to-normal ciliary beat frequency (CBF). However, the proband (II-3) and his mother (I-1) presented with an altered ciliary beat pattern (CBP) with a mainly disorganised ciliary beat, and stiff and immotile cilia (Appendix A). IF analysis showed the presence of structural markers in the ciliary axonemes of all the family members (Table 1, Appendix A). All analysed family members showed microtubular (MT) disorganisation defects with inner dynein arms (IDA) present, as observed by TEM, with the proband’s (II-3) sample being the most affected (Table 1, Figure 1d). Moreover, this sample was poorly ciliated and had short cilia; thus, cilia orientation measurement was not feasible. The mother (I-1) showed a mixed pattern, with some regions having ciliary orientation and others presenting ciliary disorientation. In the siblings (II-1 and II-2), orientation was mainly normal.

### 3.2. Identification of the Pathogenic RPGR Variant

The high-throughput PCD gene panel analysis revealed that the proband (II-3) and his brother (II-1) were hemizygous for the c.920C>A;p.(Thr307Lys) variant in the *RPGR* gene (exon 8), co-segregating with the XLRP phenotype. The mother and sister carried the same variant (Table 1, Figure 1b). This variant was absent in the gnomAD database and it was classified as variant of uncertain significance (VUS).

However, T307 residue is evolutionarily conserved (GERP = 5.2699) and the variant causes a decrease in protein stability [I-Mutant, DDG = (−1.05)]. The structural in silico analysis also showed that the mutated protein had lost the hydrogen bonds between the T307 residue and H254 and Q273 positions (Figure 2).

To further confirm the pathogenicity of the p.(Thr307Lys) RPGR variant, in vitro functional studies were performed in hTERT-RPE1 cells using RPGR_Ex1-19_ (Figure 3). Remarkably, compared to the localisation of RPGR in the wild-type form (RPGR_Ex1-19_^WT^), the mutated RPGR_Ex1-19_^T307K^ did not reach the TZ, resulting in its retention and accumulation in the cytoplasm (Figure 3a). Furthermore, analyses of the RPGR_Ex1-19_^T307K^ transfected cells resulted in significantly fewer ciliated cells (approximately 30%) than the empty GFP vector or the RPGR_Ex1-19_^WT^ construct (Figure 3b). Nonetheless, the mutant RPGR_Ex1-19_^T307K^ expression did not affect the ciliary length compared to the controls (Figure 3c).

### 3.3. RPGR Immunofluorescence Analysis in Respiratory Ciliated Cells

Our subcellular localisation analysis showed that RPGR wild-type protein is located in the TZ and was abnormally distributed in all family members (Figure 4a,b and Appendix A). The RPGR signal intensity in the TZ was significantly lower in all siblings (II-1, II-2 and II-3) compared to the control (Figure 4c). However, a higher cytoplasmic retention of RPGR was also observed in the proband (II-3) and siblings (II-1 and II-2) (Figure 4d). These results suggested that the mutant RPGR protein mostly accumulated in the cytoplasm of nasal cells. Of note, samples from both carrier female members (I-1 and II-2) seemingly showed a bimodal-like distribution of RPGR in the TZ (Appendix A).

### 3.4. Additive Effect of Modifier Genes on the PCD Phenotype

A WES analysis was conducted in all family members to analyse candidate modifier genes with a potential additive effect that could lead to PCD. As only the proband (II-3) presented with respiratory symptoms, the analysis was focused on genetic variants with a different genotype between the proband and his brother (II-1). Identified variants considered of interest are shown in Appendix A. After careful curation and considering previously published results, the *CEP290* variant c.331C>T;p.(Arg111Trp), identified in heterozygosity in the proband, sister and father, was selected as the possible modifier of the PCD phenotype in this family (Appendix A).

### 3.5. CEP290 Immunofluorescence Analysis in Respiratory Ciliated Cells

The potential effect of the *CEP290* genetic variant in the family was analysed by IF. As reported, CEP290 localised to the TZ in the control. However, it shows an altered distribution in the family members (Figure 5a,b and Appendix A). The CEP290 signal intensity in the TZ was significantly lower in the proband (II-3), brother (II-1) and mother (I-1) compared to the control sample, but no significant differences were observed in the sister (II-2) (Figure 5c). In addition, we detected no significant differences between the proband (II-3) and his brother (II-1) (Figure 5c). The percentage of CEP290 intensity in the cytoplasm showed a significantly higher accumulation in all family members compared to the control (Figure 5d). Similar to the RPGR IF results, the mother (I-1) and sister (II-2) presented as two different populations, even though *CEP290* is not located in the X chromosome (Appendix A).

## 4. Discussion

In this study, a novel *RPGR* missense variant c.920C>A;p.(Thr307Lys) was described to co-segregate with XLRP in a whole family, but with PCD respiratory symptoms only in the proband. This family case was highly interesting due to the possibility of studying the phenotypic differences between siblings (two brothers) with the same *RPGR* genotype. The proband (II-3) (Figure 1a) was referred for our consultation due to classical PCD symptoms and a PICADAR score of 6, predicting highly likely PCD (Table 1). In the study of McCray et al. [20], none of the *RPGR* patients had a PICADAR score higher than 3, although respiratory symptoms and ciliary defects were described in those cases [20]. The respiratory symptoms in the proband can be compared with those described in XLRP and PCD patients [14,15,16,17], but with neonatal symptoms. Due to these previous data and with the aim of elucidating the XLRP/PCD syndrome variability in this family, ciliary function and structure were assessed among all members. As previously reported in RP/PCD patients [14], all family members, including the proband, had normal nNO values (Table 1). Indeed, an increasing list of PCD-causing genes has been related to normal nNO values [42,43], even though low values in nNO screening tests suggest highly likely PCD [26].

The analysis of ciliary function (HSVM) in nasal brush samples showed a nearly normal CBF in all family members (Table 1), in concordance with previous data [14,20]. The CBP was nearly normal in the siblings, but altered in the proband and the mother, mainly with uncoordinated beating, stiff and immotile cilia (Table 1). These defects were described in previously reported XLRP/PCD patients [14,15,20]. Although in our family case, the most severe ciliary beat defect correlated with the respiratory symptoms diagnosed in the proband, this correlation has not been previously described in *RPGR* carriers [14].

IF analyses confirmed the presence and correct localisation of DNAH5, DNALI1, GAS8 and RSPH9 as components of different ciliary ultrastructures (Appendix A). These results are concordant with the research of other authors confirming the presence of ODA (DNAH5) and IDA (DNALI1) [14]. To our knowledge, no previous IF studies of the other ciliary structures (nexin and radial spoke head) have been reported. Thus, our results cannot be compared with others.

Our TEM study was partially informative, with some ciliary axonemal cross sections with normal ultrastructure and others with MT disorganisation defects with IDA present in all the samples. Recent published reports of *RPGR* patients described no ultrastructural defects by TEM [14], although early described data showed different axonemal structural defects in some cross sections [15]. The axonemal defects could be explained by the possible contribution of the RPGR protein in the intraflagellar transport in motile cilia, as described in the retina [16,21]. The proband’s (II-3) sample was the most affected, in concordance with more severe CBP defects in HSVM (Table 1, Figure 1d). Furthermore, as *RPGR* patients were previously described to have orientation defects [14,20], we aimed to study this in the family. Unfortunately, the proband’s sample was poorly ciliated; thus, cilia orientation measurement was not feasible. In the mother, heterozygous for the *RPGR* variant, central pairs were orientated in some sections and disorientated in others, consistent with random X-chromosome inactivation. On the contrary, siblings II-1 and II-2 showed a mostly normal orientation, correlating with the absence of respiratory symptoms.

In nasal brush control samples, RPGR is consistently localised to the TZ of motile cilia, as it occurs in primary and neurosensorial cilia (Figure 4a,b) [22,24]. Unfortunately, our attempt to study the RPGR exact localisation by TEM immunogold labelling was unsuccessful. The localisation of RPGR in the TZ also corroborates its possible role in the planar cell polarity (PCP) pathway, which determines the orientation of basal bodies and ciliary beat coordination [14,44,45]. However, further studies are necessary to study the exact function of RPGR in the correct establishment of ciliary orientation. The IF analysis significantly showed a considerably reduced RPGR intensity in the TZ of the proband and siblings, and a higher RPGR cytoplasmic intensity suggested cytoplasmic accumulation (Figure 4c,d). These results agree with our in vitro functional analyses in retinal cells (Figure 3a). It is noteworthy that the mother and sister displayed two different subpopulations of cells, with some cells having a higher RPGR intensity and TZ localisation (similar to the control), and some cells with an aberrant RPGR localization (Appendix A). These results could be due to the random X-chromosome inactivation in female carriers.

The novel *RPGR* variant c.920C>A;p.(Thr307Lys) was classified as VUS according to ACMG criteria, and co-segregated with the XLRP phenotype. The in silico predictions (Figure 2) and in vitro functional studies (Figure 3) confirmed the pathogenicity of the variant. Note that the mother carried the variant and suffered RP, but not the sister. It is not infrequent that some female carriers of *RPGR* mutations show an altered retinal phenotype, which is attributed to the randomisation of X-chromosome inactivation during retinal development [15,46]. We also must not disregard the fact that the sister may suffer from RP in time, as RP is a degenerative disease with a progressive evolution [6].

Overall, the results indicate that the *RPGR* c.920C>A variant also produces a protein defect in respiratory motile cilia. Nevertheless, we cannot explain the presence of respiratory symptoms solely in the proband, sharing the same hemizygous genotype with his brother. Additionally, phenotypic incomplete penetrance of the PCD symptoms in XLRP mutations has been described [13,14,15]. Therefore, other genetic or environmental factors may be involved in the phenotypic variability in this family. Consequently, we performed a WES study in this family with a particular focus on the proband (II-3) versus his brother (II-1). We searched candidate variants exclusively present in the proband or carried in different genotypes between both brothers, pointing to a missense variant in *CEP290* (Appendix A). The centrosomal protein 290 (CEP290) is an evolutionarily conserved TZ protein essential for initiating TZ formation and early ciliary membrane formation during ciliogenesis in *Drosophila* [47]. Heterozygous *CEP290* variants in *RPGR*-knock-out male mice have been described to cause a worse retinal degeneration progression, being a genetic modifier of XLRP severity [48]. *CEP290* has been previously described as affecting the overlapping phenotype of various ciliopathies caused by mutations in other ciliary genes [49,50,51]. However, the human RPGR physically interacts with the C-terminal domain of CEP290, thus forming a complex necessary for TZ morphology and probably facilitating ciliary trafficking [48]. Interestingly, Papon et al. [52] described that homozygous *CEP290* patients with Leber congenital amaurosis (LCA) frequently presented with ultrastructural defects in their respiratory cilia [52]. Considering all these data, the c.331C>T;p.(Arg111Trp) variant in *CEP290*, carried only in heterozygosis in the proband and his sister and father, was considered the most probable genetic modifier in the presented family case. To our knowledge, this is the first time that a modifier gene has been postulated as a possible explanation for PCD symptom variability.

Considering our interest in studying the effect of the *CEP290* variant in the respiratory cilia and explaining the possible cause of differences in phenotypes, we focused on the analysis of the CEP290 expression in the respiratory ciliated tissue from the family members. These sublocalisation studies in respiratory epithelia confirmed that CEP290 was present in the TZ (Figure 5a,b), but its expression was significantly reduced in the proband (II-3) and his brother (II-1). However, it accumulates in the cytoplasm, similar to RPGR. No significant differences were observed between the proband (II-3) and his brother (II-1), although only the proband carries the *CEP290* variant (Figure 5c). Thus, the similarity in the CEP290 localisation between brothers does not demonstrate the effect of the *CEP290* genetic variant, but instead reinforces the effect of the mutated RPGR and its interaction with CEP290, causing the mislocalisation of the latter. The mislocalisation of CEP290 on the *RPGR* mutation background is an interesting observation. The standard mislocalised proteins in PCD are not altered in these patients, which suggests that different ciliary protein complexes might be disorganised in PCD patients depending on the genotype. Considering our results and all the previously reported data in the literature, the *CEP290* gene may be a potential modifier gene for the effect of *RPGR* mutations on PCD patients. High-throughput sequencing techniques allow the analysis of multiple causative and candidate genes. Therefore, we propose that *CEP290* might be considered as a causal and modifier gene in motile ciliopathies. Further studies of the specific contribution of CEP290 and RPGR and their interaction in motile cilia should be performed.

## 5. Conclusions

We present an exhaustive motile ciliary characterisation in a XLRP family with two siblings hemizygous for a missense variant in *RPGR*, but with PCD symptoms only present in one. The pathogenicity of the *RPGR* variant is confirmed but does not explain the respiratory symptoms by itself. Finally, the *CEP290* gene may be postulated as a potential modifier of respiratory symptoms in patients carrying *RPGR* mutations. 

## Figures and Tables

**Figure 1 cells-13-00524-f001:**
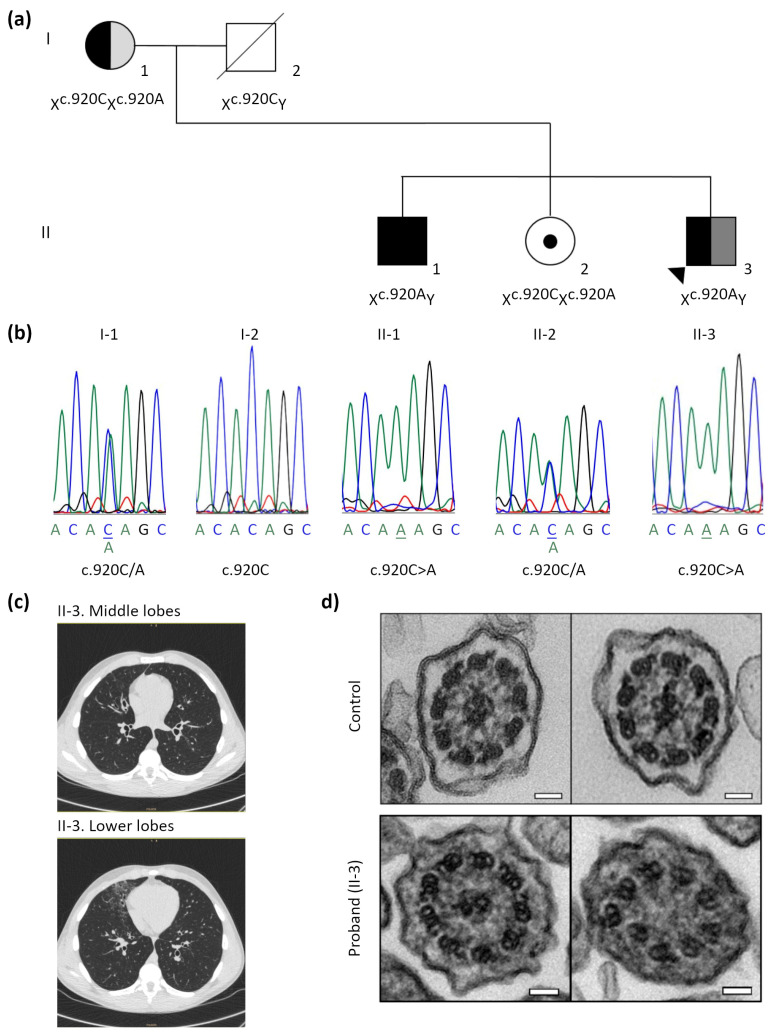
X-linked retinitis pigmentosa (XLRP) family pedigree and transmission electron microscopy (TEM) analysis of nasal brush samples. (**a**) Pedigree of the XLRP family with the proband (II-3, arrow) presenting with retinitis pigmentosa (black) and primary ciliary dyskinesia respiratory symptoms (dark grey); the mother (I-1) presented with asthma (light grey). (**b**) Electropherogram of all family members showing the *RPGR* c.920C>A variant in hemizygous state in the proband (II-3) and brother (II-1). The mother (I-1) and sister (II-3) were carriers of this variant. The locus is marked with an asterisk. (**c**) Computed axial tomography scan of the proband (II-3) showing bronchiectasis in the middle and lower lobes. (**d**) Cross sections of control and RPGR proband (II-3) samples of airway ciliary axoneme by TEM; in the proband sample (below), normal cross sections with 9 + 2 motile cilia axoneme (left) and microtubular disorganisation with inner dynein arms present (right) were observed. Scale bar, 50 nm.

**Figure 2 cells-13-00524-f002:**
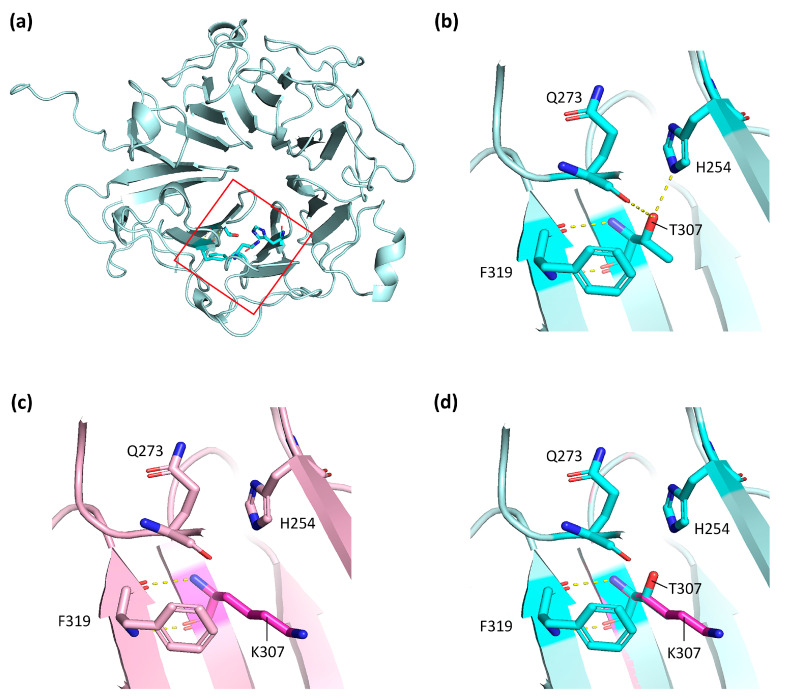
In silico studies of c.920C>A;p.(Thr307Lys) variant in the *RPGR* gene. (**a**) Tertiary structure of RPGR^WT^ RCC1 domain from AlphaFold. (**b**) Zoom in of the RPGR^WT^ RCC1 domain at the T307 position. The dotted yellow lines represent the hydrogen bonds that T307 amino acid establishes with other protein residues. (**c**) Zoom in of the RPGR^T307K^ RCC1 domain at the K307 position, modelled with PyMOL. Note that the hydrogen bonds with H254 and Q273 are lost. (**d**) Merging of RPGR^WT^ and RPGR^T307K^ RCC1 domain structures to show the shift in the beta-sheet where the 307 residue lies.

**Figure 3 cells-13-00524-f003:**
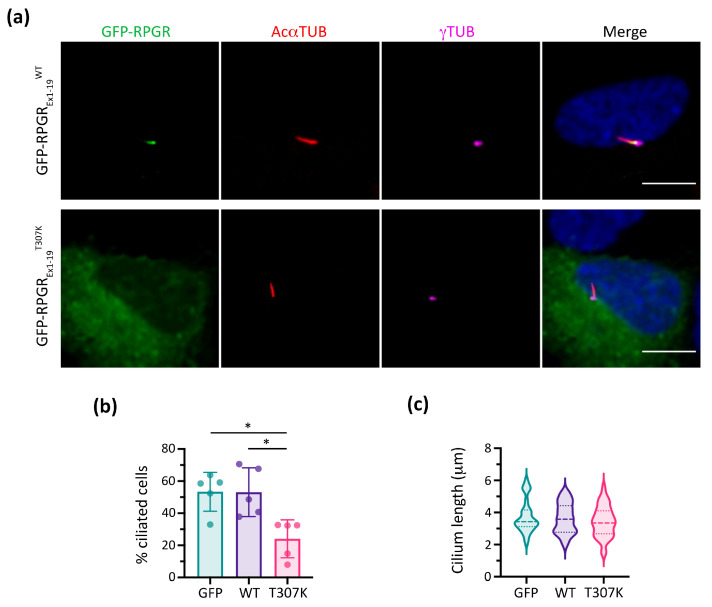
In vitro functional studies of p.(Thr307Lys) *RPGR* variant in human retinal pigment epithelium (hTERT-RPE1) cells. (**a**) Cilium localisation of transfected GFP–RPGR_Ex1-19_^WT^ and mislocalisation of transfected GFP–RPGR_Ex1-19_^T307K^ in ciliated hTERT-RPE1 cells. Scale bar, 10 μm. (**b**) Percentage of ciliated cells among the GFP-positive hTERT-RPE1 cells (Kruskal–Wallis test, * *p* < 0.05, *n* = 5). (**c**) Measurement of ciliary lengths in GFP-positive ciliated hTERT-RPE1 cells (one-way ANOVA, *n* = 24–39 from three independent replicates). AcαTUB: acetylated α-tubulin; γTUB: γ-tubulin.

**Figure 4 cells-13-00524-f004:**
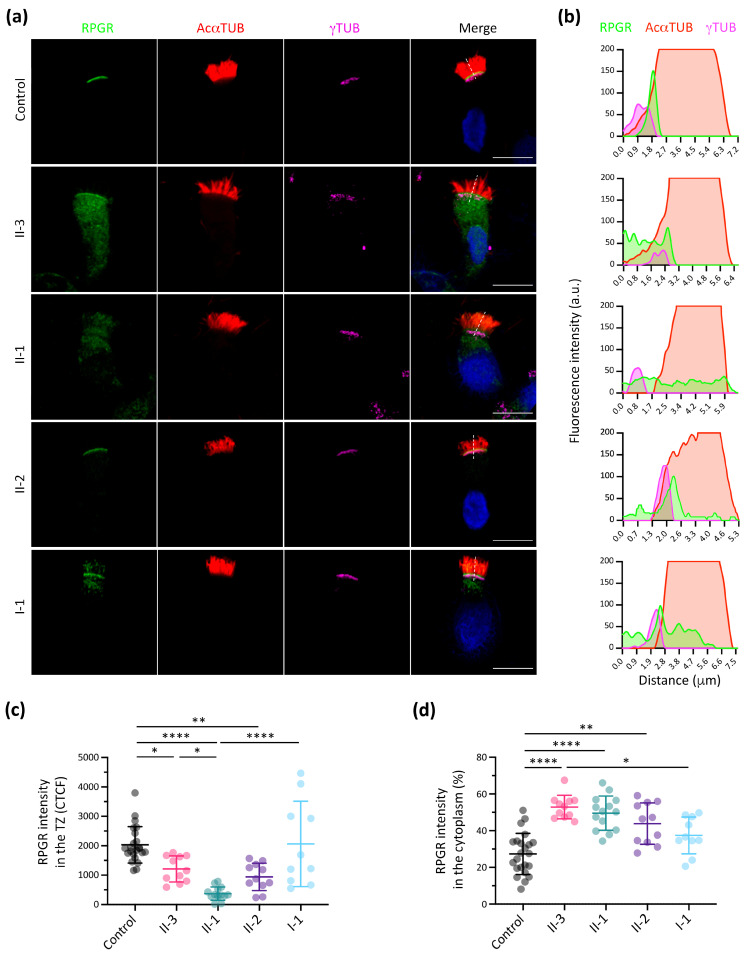
Immunofluorescence of RPGR protein in nasal brushing samples. (**a**) Representative images of RPGR subcellular localisation in individual respiratory epithelial cells from nasal brushing biopsies. The RPGR protein locates at the transition zone (TZ), distal to basal bodies but proximal to ciliary axoneme [labelled by γ-tubulin (γTUB, magenta) and acetylated α-tubulin (AcαTUB, red), respectively]. Nuclei were stained with DAPI (blue). Scale bar, 10 μm. (**b**) Plot profile of the distribution of RPGR (green) along the discontinuous white lines compared with acetylated α-tubulin (red) and γ-tubulin (magenta) for each image on the left. a.u., arbitrary units. (**c**) Quantification of the RPGR intensity signal at the TZ. (**d**) Cytoplasmic percentage of RPGR intensity signal in respiratory epithelial cells from nasal brushing biopsies. Kruskal–Wallis test; *n* = 10–14; * *p* < 0.05, ** *p* < 0.01, **** *p* < 0.0001.

**Figure 5 cells-13-00524-f005:**
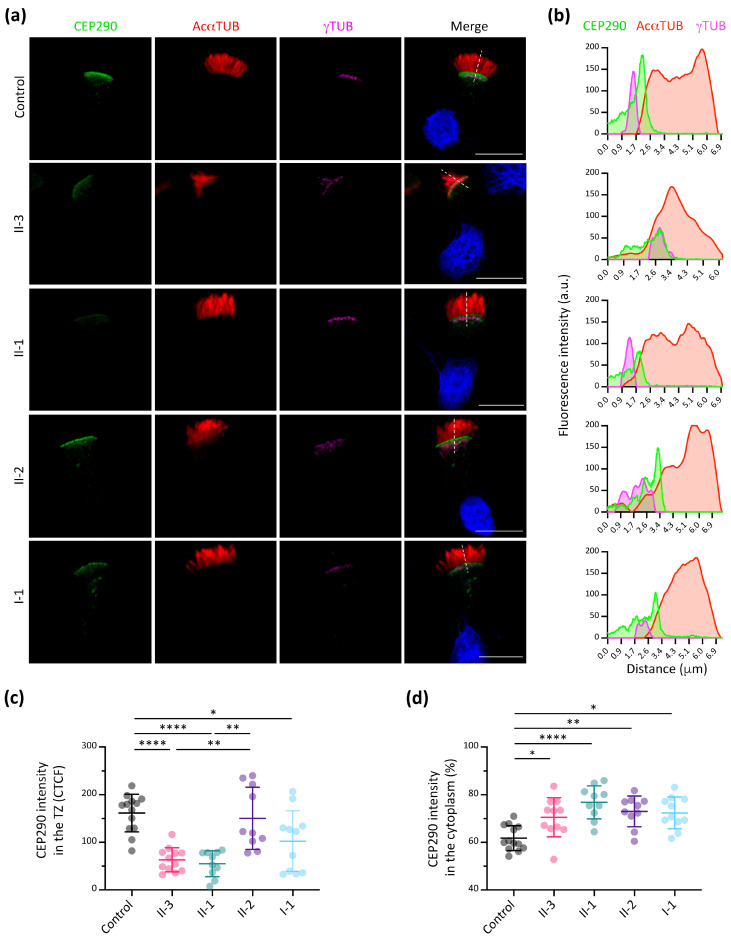
Immunofluorescence of CEP290 protein in nasal brushing samples. (**a**) Representative images of CEP290 subcellular localisation in respiratory epithelial cells from nasal brushing biopsies. The CEP290 protein locates at the transition zone (TZ), distal to basal bodies but proximal to ciliary axoneme [labelled by γ-tubulin (γTUB, magenta) and acetylated α-tubulin (AcαTUB, red), respectively]. Nuclei were stained with DAPI (blue). Scale bar, 10 μm. (**b**) Plot profile of the distribution of CEP290 (green) along the discontinuous white lines in cilia compared with acetylated α-tubulin (red) and γ-tubulin (magenta) for each image on the left. a.u., arbitrary units. (**c**) CEP290 intensity signal at the TZ. (**d**) Cytoplasmic percentage of CEP290 intensity signal of respiratory epithelial cells from nasal brushing biopsies. Kruskal–Wallis test; *n* = 10–13; * *p* < 0.05, ** *p* < 0.01, **** *p* < 0.0001.

**Table 1 cells-13-00524-t001:** Clinical manifestations and PCD diagnostic tests.

Subjects	I-1	I-2	II-1	II-2	II-3 (Proband)
**Clinical manifestation**	RP + asthma	Healthy	RP	RP carrier	RP + PCD respiratory symptoms
**PICADAR score**	3	NA	2	1	6
**nNO (nL/min)**	314.2	NA	328.2	533.8	107.8
**HSVM ^1^**	8.46 Hz, 45.7% DC (D and S)	NA	11.68 Hz, 23% DC (D and S)	11.74 Hz, 26% DC (D and S)	8.67 Hz, 95.4% DC (D, S and I)
**IF ^2^**	All markers present	NA	All markers present	All markers present	All markers present
**TEM**	33% MT disorganisation with IDA present	NA	16% MT disorganisation with IDA present	17% MT disorganisation with IDA present	25% MT disorganisation with IDA present
**Genetics**	*RPGR*:c.920C>A het.		*RPGR*:c.920C>A hem.	*RPGR*:c.920C>A het.	*RPGR*:c.920C>A hem.

RP: retinitis pigmentosa; PCD: primary ciliary dyskinesia; PICADAR: PrImary CiliARy DyskinesiA Rule; nNO: nasal nitric oxide; HSVM: high-speed video-microscopy; IF: immunofluorescence; TEM: transmission electron microscopy; NA: not available; DC: dyskinetic ciliated cells; D: disorganised ciliary beat; S: stiff ciliary beat; I: immotile cilia; MT: microtubular; IDA: inner dynein arms; het: heterozygous patient; hem: hemizygous. ^1^ The high-speed video-microscopy analysis of nasal brushing samples consisted of ciliary beat frequency, percentage of analysed cells with ciliary beat pattern defect and the specific pattern defect observed (indicated in parentheses). ^2^ Structural ciliary defects were assessed by immunofluorescence, using antibodies against the ciliary proteins DNAH5, DNALI1, GAS8 and RSPH9.

## Data Availability

The original contributions presented in the study are included in the article/Appendix A; further enquiries can be directed to the corresponding author/s.

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
