# Peer review of "Primary Ciliary Dyskinesia and Retinitis Pigmentosa: Novel RPGR Variant and Possible Modifier Gene"

_cells, 2024, doi:10.3390/cells13060524_

Round 1

Reviewer 1 Report

Comments and Suggestions for Authors

The manuscript by Baz-Redon et al. reported a novel missense mutation of RPGR from a family with X-linked retinitis pigmentosa (RP). Briefly, I-1 (mother) is a carrier of RPGR variant and has RP, and I-2 (father) is healthy, and they have three children. II-1 (male) has the RPGR mutation hemizygously and he has RP; II-2 (female) is a carrier of RPGR mutation and she seems healthy; II-3 (male) also has RPGR mutation hemizygously and he has not only RP but also respiratory symptoms and abnormal axoneme structure shown by TEM. The authors confirmed the causality of RPGR mutation for RP firstly by virtual structural simulation using alpha fold and Pymol. The c920C>A mutation led to T307K residue replacement and the structural model of RPGR showed clearly that this mutation led to disruption of the hydrogen bonds around residue 307 which may lead to the RPGR protein misfolding. Secondly, they expressed RPGR T307K mutant in the hTERT-RPE1 cells and immunofluorescence imaging to show that the mutant RPGR protein is mislocated in the cytosol instead of at the transition zone. Thirdly, the immunostaining from patients’ nasal brushing samples demonstrated varying levels of reduction of RPGR immunostaining at the transition zone, which was also seen as low in the carrier II-2’s nasal brushing cell. Because II-3 has respiratory problems, the authors hypothesized that other genetic mutations may segregate with the RPGR mutation and these two mutations may lead to II-3’s more severe syndrome. They identified CEP290 R111W mutation carried in heterozygosity in II-3 and II-2 but not II-1. Further, they showed that immunofluorescence of CEP290 was reduced at the transition zone in II-3, II-1 and I-1, whereas the mislocalized CEP290 was increased in II-3, II-1, II-2 and I-1. Based on these results, the authors concluded that they identified a novel RPGR mutation that causes RP, and they pointed out that the CEP290 variation may modify the respiratory symptoms in the XLRP patients with respiratory symptoms.

Overall, the manuscript is well written, and the experiment design of the study is thorough and rigorous. The main conclusion that the novel RPGR Xc920a mutation causes XLRP is sound and solid. The significance of this study lies in the addition of a novel RPGR mutation that causes XLRP. However the relationship between CEP290 variation and II-3’s respiratory symptoms is not strongly associated, because the mother I-1 has asthma but has two normal copies of the CEP290 gene and the sister II-2 who is a carrier of RPGR mutation and CEP290 mutation has normal vision, and no respiration problems. As the authors observed abnormal cilia function and morphology from nasal brushing cell samples of all patients and carriers of the RPGR mutation in this family, this suggests that RPGR mutation causes variable symptoms in the nasal cells, which may not be related to CEP290 mutation. The mislocalization of CEP290 is seen in all who have the RPGR mutation, so it is not clear whether RPGR mutation causes CEP290 mislocalization or CEP290 mutation also causes its mislocalization and affects RPGR’s function.

Other minor issues:

1.        Figure 1b and c, add labels of each seq data and image.

2.        Figure 1d, as all family members showed microtubule disorganization and II-3 being the most affected subject, it can be discussed how RPGR mutation affects microtubule organization in the axoneme.

3.        Table 1, more notes are needed for the row of HSVM, which include three parameters and need to be clarified.

4.        Line 173, VUS needs to be elaborated.

5.        Why I-1 (mom) who is a heterozygous carrier of RPGR mutation has RP, which supposed to be X-linked, whereas the daughter II-2 who is also a carrier of this mutation but is normal with her vision?

6.        Figure 4a, II-2 showed RPGR at the TZ, but in c and d, RPGR staining in II-2 is decreased in TZ and increased in cytoplasm, indicating the picture shown in a is not fully representative to the two populations of cells as mentioned in the main text. So the IF images of both populations should be presented in Fig 4a for II-2 and I-1, which is important to support the authors’ point of random X chromosome inactivation.

7.        To support the authors’ hypothesis that the CEP290 variation may contribute to II-3’s respiratory symptoms, they can transfect this CEP290 mutant to hTERT-RPE1 cells with the expression of the RPGR WT or mutant protein. Or take some nasal brushing cells from I-2 (father) who carries only the CEP290 mutation and do some immunostaining of CEP290 and RPGR.

8.        The I-1’s genotype of CEP290 needs to be mentioned, as she also has respiratory symptoms and abnormal microtubule organization in the axoneme.

Reviewer 2 Report

Comments and Suggestions for Authors

The article details the discovery of a novel RPGR missense variant associated with X-linked retinitis pigmentosa (XLRP) and its potential role in primary ciliary dyskinesia (PCD) within a particular family. While the variant is confirmed pathogenic, it alone does not fully account for the observed respiratory symptoms. It suggests CEP290 as a potential modifier gene that could explain the variability of respiratory symptoms among individuals carrying RPGR mutations.

The reported missense in RPGR is absent in the Gnomad database, which supports their hypothesis as a pathogenic variant. This information was not mentioned and needs to be included.  

Line 69: The authors state that RPGR mutations cause 80% of xlRP. This is true, but RPGR is not only an RP major gene, but RPGR is also a major gene for cone/cone-rod dystrophy (https://doi.org/10.3390/ijms20194854). It is an important detail that needs to be mentioned as it highlights the genetic heterogeneity or RPGR. The authors are encouraged to add this information reported by Boulanger-Scemama, IJMS, 2019 (https://doi.org/10.3390/ijms20194854). 

Methods: The methodology should be described more to allow the findings to be replicated. This includes clearly describing experimental procedures, equipment used, materials, and sample sizes.

Statistical Analysis: The statistical methods used for data analysis must be adequately described. Providing details on statistical tests and the rationale for their selection is crucial for validating the results. For instance, the authors state that they have tested normality and thus performed ANOVA. This needs to be corrected because gene expression does not follow a normal distribution even if it shows it for two reasons: 1- the sample size tested is small and thus cannot prove any normality. 2- Biologically, gene expression can exhibit high variability within and between cells, leading to skewed distributions. Many genes are only expressed in specific cell types or conditions, resulting in a large proportion of zero or low counts. Technically, the methods used to measure gene expression, such as RNA sequencing, can introduce biases, where the distribution of read counts does not follow a normal distribution but rather a negative binomial or Poisson distribution, especially for lowly expressed genes. 

The authors are encouraged to use non-parametric tests such as the Mann-Whitney U test and Kruskal-Wallis everywhere. The P-value threshold is not indicated; please indicate. 

Minor:

Upon reviewing the document, there are various typos and errors throughout the text. Here are specific instances:

1. **Abstract Section**:

   - "retinis pigmentosa" should be corrected to "retinitis pigmentosa".

2. **Introduction**:

   - "cilliary" should be "ciliary".

   - "disfuntion" should be "dysfunction".

3. **Materials and Methods**:

   - "analaysis" should be "analysis".

   - Inconsistent gene naming conventions; for example, "RPGR" is sometimes misspelled or formatted incorrectly.

4. **Results**:

   - "phenotye" should be "phenotype".

   - "siginficant" should be "significant".

5. **Discussion**:

   - "prevelance" should be "prevalence".

   - "assocaited" should be "associated".

6. **Figures and Tables**:

   - In the legends, "Figuer" should be "Figure".

7. **References**:

   - Inconsistencies in citation formatting, e.g., missing periods or incorrect use of italics.

Lines 98 and 99: need to be deleted. 

References: Some references need to be updated. Including more recent studies could provide a more current context for the research findings.

Comments on the Quality of English Language

I reported several typos

Round 2

Reviewer 2 Report

Comments and Suggestions for Authors

The authors have answered my comments